# Genetic Diversity of Ancient *Camellia sinensis* (L.) O.Kuntze in Sandu County of Guizhou Province in China

Yichen Zhao [1,*] , Runying Wang [2], Qing Liu [2], Xuan Dong [1] and De-Gang Zhao [1,3]

1    College of Tea Sciences, The Key Laboratory of Plant Resources Conservation and Germplasm Innovation in Mountainous Region (Ministry of Education), Guizhou University, Guiyang 550025, China; xdong@gzu.edu.cn (X.D.); dgzhao@gzu.edu.cn (D.-G.Z.)
2    College of Life Sciences, Guizhou University, Guiyang 550025, China; gs.runyingwang19@gzu.edu.cn (R.W.); lq20210603@163.com (Q.L.)
3    Guizhou Academy of Agricultural Science, Guiyang 550006, China
*    Correspondence: correspondence: yczhao@gzu.edu.cn

**Abstract:** The ancient tea plant germplasm is an important resource for breeding new tea plant varieties and has great economic value. However, due to man-made and natural disturbances, it has become endangered. In order to have a better management of the conserved tea plant germplasm, it is a requirement to understand the genetic and phenotypic diversity. The aim of this study was to evaluate the genetic and phenotypic diversity of 145 ancient tea plant germplasm resources from five populations in Sandu County of Guizhou province in China. To explore the population genetics of tea plant, we successfully identified 15 simple sequence repeat (SSR) markers, which were highly polymorphic. Additionally, we applied traditional phenotypic methods to evaluate the tea plant diversity. The results suggested that the genetic and phenotypic diversity were relatively high. A total of 96 alleles were identified, and the mean polymorphic information content (PIC) value was found to be 0.66. The analysis of molecular variance (AMOVA) showed that genetic variation within the populations was greater than among the populations. Overall, our results are the valuable baseline data in developing more efficient management and breeding plans for one of the most popular non-alcoholic beverage crops, the tea plant species.

**Keywords:** ancient tea plant; phenotypic characters; genetic diversity; SSR markers

## 1. Introduction

Ancient tea plant (*Camellia sinensis* (L.) O.Kuntze) germplasm refers to the tea tree that is more than 100 years old [1]. The ancient tea plant germplasm does not require chemical control and additives, but it is rich in nutrients. Thus, tea made from the ancient tea plant germplasm is more popular [2]. Therefore, it has significant scientific, social, economic, and cultural benefits, and is an important agricultural cultural heritage in the world [3].

Guizhou is one of the origin sites of tea plants [4]. Recent studies suggest that Sandu Aquarium Autonomous County in Guizhou, China is highly diverse in the ancient tea plant germplasm. This county is located at high altitudes in Yunnan–Guizhou Plateau, with suitable ecological and climate conditions and less environmental pollution. Thus, this region provides a suitable environment for the ancient tea plant germplasm. The exploration of the genetic diversity of this germplasm may facilitate the effective protection and utilization of these tea plant resources [5]. Tea plants are self-incompatible, which promotes relatively highly heterozygous progenies and produces rich genetic variations. Therefore, revealing the interspecific differences and genetic diversity at the genome level is important for the understanding of the genetic mechanism of tea plant growth and development [6]. The importance of the genetic diversity analysis of genetic resources for protection and breed selection has been well recognized [7]. The ancient tea plant germplasm in Sandu of Guizhou survived hundreds or even thousands of years. After

a long period of evolution, the ancient tea plant germplasms have formed unique phenotypic, physiological and biochemical characteristics [8]. In addition, the ancient tea plant germplasm has rich genetic diversity, including almost all kinds of primitive and evolutionary types, which are essential materials to explore the origin and evolution of the tea plant germplasm [9]. Therefore, the genetic diversity of the ancient tea plant germplasm was substantially analyzed. This can be routinely assessed by means of several techniques, such as (i) phenotypic [10], and (ii) molecular marker analysis [11].

Phenotypic research methods are based on visually accessible traits, such as leaf color, leaf shape, and growth habits. The research methods based on morphological analysis are easy to implement and are commonly used in the studies of biological genetic diversity [12]. However, morphological traits are easily affected by the environment and climate [13–16]. Nowadays, the most effective markers for the detection of the genetic diversity of plants are DNA molecular markers [17]. Unlike morphological data, molecular approaches provide extensive information that is not influenced by environmental factors [18]. Therefore, molecular markers have been proved to be effective tools in the study of genetic variability and the assessment of interrelationships among different genotypes [19]. Microsatellites, also known as simple sequence repeats (SSRs), are molecular markers, which were widely used in genetic linkage mapping, genetic diversity detection, and molecular-assisted breeding. SSR markers have the advantages of codominance, simple operation, good universality, and high repeatability. Therefore, the SSR markers an essential agricultural industry standard for the identification of wheat [20], watermelon [21], pepper [22], *Toona sinensis* (Juss.) M.Roem. [23], rice [24], and many other crop varieties.

Due to global warming and increasing disturbances (e.g., fire, insects) [25–27], it is necessary to have a better understanding of the genetic diversity of the ancient tea plant germplasm and improve the existing protecting and management strategies. Thus, in this project, an assessment of the phenotypic traits and SSR markers was made to analyze the genetic diversity of the ancient tea plant germplasm. To investigate the population genetic of the ancient tea plant germplasm, we have developed 15 microsatellite markers. Finally, in this study, we provided a theoretical basis for the effective collection, conservation, and utilization of germplasm resources, and promoted further research of the ancient tea plant germplasm.

## 2. Materials and Methods

### 2.1. Plant Materials

The plant materials of tea plant for genetic and morphometric analyses were collected from four plots (i.e., Guqi Village, Landong Village, Yangmeng Village, and Zenya Village) and five populations in May 2019 in the Sandu Aquatic Autonomous County of Guizhou province, China (Figure 1). Their geographical locations and the number of samples are provided in Table 1. The sampled ancient tea plant germplasm individuals were 30–50 cm in diameter, about 10 m in height and over 100 years old. The tender buds of each ancient tea plant germplasm with one bud and one leaf or one bud and two leaves were picked and placed in a self-sealed bag, and then filled with dried silica gel. The fifth mature and fully developed leaf of each ancient tea plant germplasm was collected and brought back to the laboratory for morphometric analysis.

**Table 1.** List of studied populations of ancient tea plant germplasm used for genetic and phenotypic analysis, as well as their geographical information.

| Location | Tree Form | Longitude | Latitude | Mean Altitude (m) | Sample Size |
|---|---|---|---|---|---|
| Guqi Village | Bush | 107°59′34″ E | 25°44′34″ N | 689.1 | 22 |
| Landong Village | Bush | 108°0′20″ E | 25°42′40″ N | 988.7 | 15 |
| Yangmeng Village | Arbor/Bush | 107°50′11″ E | 25°45′8″ N | 884.0 | 10 |
| Landong Village | Arbor | 108°0′30″ E | 25°42′30″ N | 905.2 | 44 |
| Zenya Village | Arbor | 108°9′37″ E | 25°59′36″ N | 1299.6 | 54 |



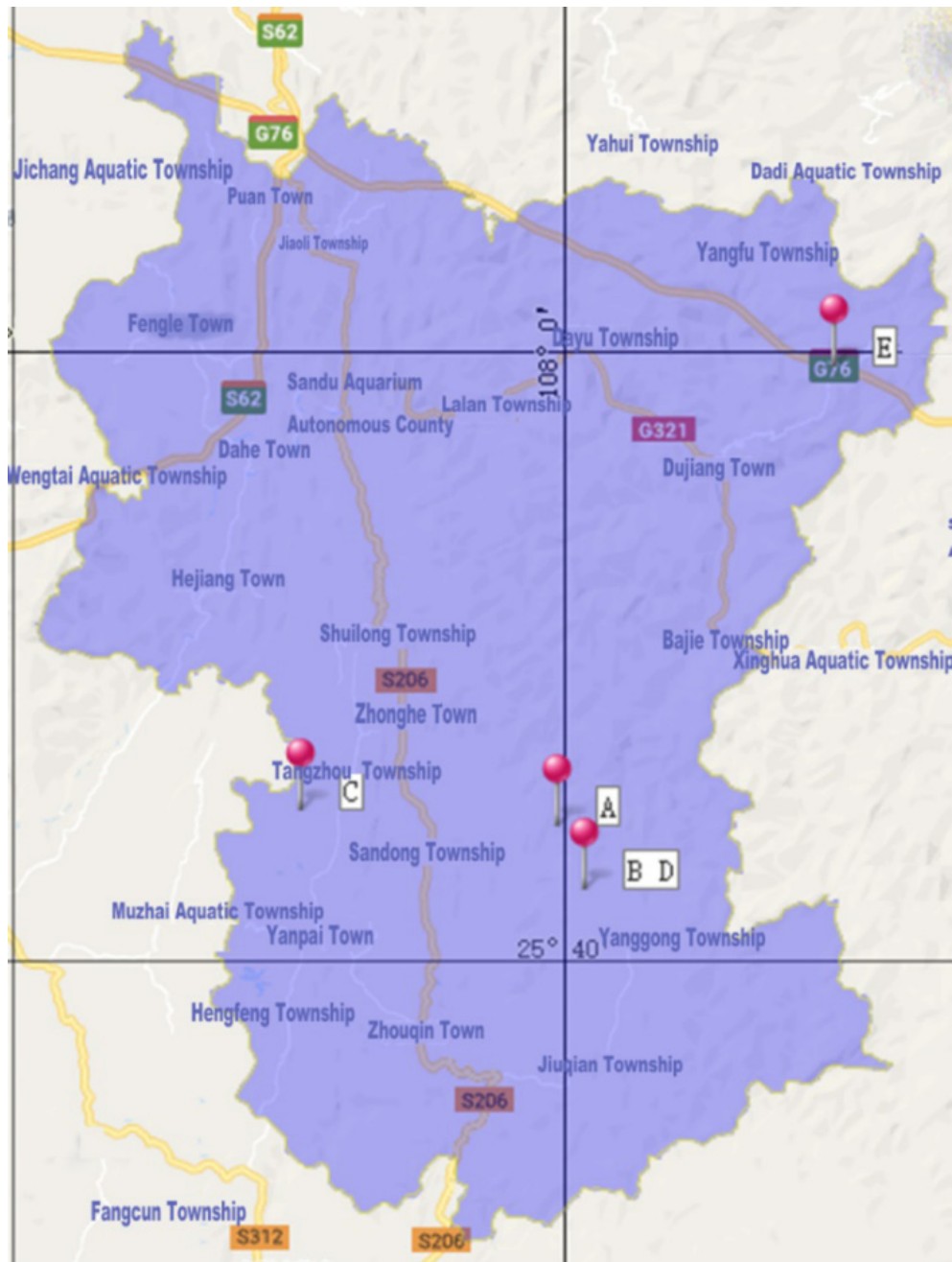

**Figure 1.** The locations of studied populations of ancient tea plant germplasm used for genetic and phenotypic analysis. A: shrubby ancient tea plant germplasm in Guqi Village; B: shrubby ancient tea plant germplasm in Landong Village; C: arboreal ancient tea plant germplasm in Yangmeng Village; D: arboreal ancient tea plant germplasm in Landong Village; E: arboreal ancient tea plant germplasm in Zenya Village.

*2.2. Phenotypic Data Analysis*

Phenotypic characters of the ancient tea plant germplasm were recorded immediately after collecting [28]. All data sets were collected by a single person to minimize the visual errors for certain phenotypic characters. The phenotypic characters were categorized as quantitative and qualitative characters. The quantitative characters comprised leaf veins logarithmic, tree height, leaf teeth logarithmic, leaf area, leaf length, and leaf width [29]. The values of the quantitative characters were calculated following the instruction of direct measurement of collected samples. The qualitative characters comprised tree form, leaf

texture, leaf apex, leaf shape, leaf size, leaf tooth depth, and leaf color. The "Description specification and data standard of tea germplasm resources" was used to assign the qualitative characters of these collected samples (Table S1) [30]. After data standardization, Microsoft Excel 2010 was used for statistical analysis of the experimental data, and the SPSS 19.0 software was used for diversity analysis, principal component analysis (PCA), as well as cluster analysis [31,32]. The PCA plot was constructed to identify the trait that efficiently differentiated the ancient tea plant germplasm populations by using the R program [33].

### 2.3. DNA Extraction, SSR Marker Development and PCR Analysis

Total genomic DNA of the ancient tea plant germplasm was extracted using a modified cetyl trimethylammonium bromide (CTAB) method [34]. Reduced representation sequencing was performed and then the novel genomic simple sequence repeat markers were obtained. The primers were designed according to the microsatellite repeat sequences and used to amplify the experimental samples. Finally, the appropriate primers were selected according to the amplification results. The compounds in the PCR mixture for amplification comprised 1.5 µL DNA (2 ng/µL), 10.72 µL ddH$_2$O, 1.5 µL 10× buffer (Mg$^{2+}$), 0.6 µL dNTP (2.5 mmol/L), 0.08 µL Easy Taq (5 U/µL), 0.3 µL primer F (10 µmol/L), and 0.3 µL primer R (10 µmol/L). The PCR program had the following two stages: in the first stage, pre-denaturation was performed at 94 °C for 4 min, denaturation at 94 °C for 40 s, annealing at Tx + 8 °C for 30 s (Tx is the optimal annealing temperature for the primer), and extension at 72 °C for 90 s, for 16 cycles (temperature in each cycle was reduced by 0.5 °C). In the second stage, the annealing temperature was reduced to Tx, denaturation was performed at 94 °C for 40 s, annealing at Tx for 30 s, extension at 72 °C for 90 s, for 24 cycles, and final extension was performed at 72 °C for 12 min and stored at 12 °C. The SSR-PCR products were detected by capillary electrophoresis [35]. The capillary electrophoresis apparatus was developed and produced by AATI Corporation in America. The SSR markers were performed in triplicate.

### 2.4. Genetic Diversity Analysis Using SSR Markers

PopGene 32 was used to compute the observed number of alleles (Na), the effective number of alleles (Ne), the Nei gene diversity index (H), the Shannon diversity information index (I), the genetic distance, and the genetic consistency [36]. The polymorphic information content (PIC) values were calculated for each SSR locus. The analysis of molecular variance (AMOVA) was performed to analyze genetic variation among individuals by the GenAlex software version 6.5, testing Fst by 9999 random permutations [37]. A model-based (Bayesian) clustering was performed to evaluate the population structure using the STRUCTURE software and the program was set up and executed as reported by Mercati et al. [38]. The NTSYS PC-2.1 software was used for cluster analysis following the unweighted average method (UPGMA), which was used to construct the cluster diagram.

## 3. Results

### 3.1. Diversity Analysis of Phenotypic Characters

Phenotypic diversity analysis was conducted on seven qualitative and six quantitative characters of these 145 ancient tea plant germplasm resources. These results are reported in Tables S2 and S3, with the coefficients of variation ranging from 17.76% to 60.37%. The coefficient of variation of tree height was the highest, which was 60.37%. The variation coefficients of nine phenotypic characters, except for leaf texture, leaf veins logarithmic, leaf length, and leaf width, were higher than 25%, which indicated that the phenotypic characters of the studied material had a high degree of variation. The diversity index of the 13 phenotypic traits ranged from 0.55 to 2.74, and leaf area was the highest (2.74), and leaf texture was the lowest (0.55). The diversity indices for all the phenotypic characters, except leaf texture, were higher than 0.75. The results showed that the phenotypic diversity of the 145 experimental materials of the ancient tea plant germplasm was relatively high, which provided a wide range of parents for breeding new varieties.

The variation coefficient and genetic diversity indices of the ancient tea plant germplasm also varied significantly among the different populations (Tables S4 and S5). The tree form of the ancient tea germplasm collected from the Landong and Guqi villages were shrubs, and thus the variation coefficient and genetic diversity index of tree shape were not included in the study. The leaf texture from the Yangmeng and Landong villages (bush) was relatively hard, so the variation coefficient and diversity index of leaf texture in these two populations were excluded. The variation coefficient of the phenotypic traits of the five tea plant germplasm populations ranged from 23.92% to 27.39% (Table S4), indicating that there was a high degree of variation among the phenotypic traits of the ancient tea plant germplasm populations in this experiment. The diversity indices of the phenotypic traits of the five tea plant germplasm populations ranged from 1.37 to 1.87 (Table S5), indicating that the phenotypic diversity of the ancient tea plant germplasm populations in the experiment was relatively high. Finally, we found that among the five populations, the variation coefficient and the diversity index of the ancient tea plant germplasm in Zenya Village were the highest. The genetic diversity of the ancient tea plant germplasm population in Zenya village was the most abundant.

### 3.2. Principal Component Analysis of Phenotypic Characters

To examine the role of each character in the phenotypic diversity of the ancient tea plant germplasm, 13 phenotypic characters were considered for a principal component analysis [39]. The contribution rate of the first six components was 85.48%, indicating that the features of the 13 phenotypic characters can be reflected (Table S6) [40]. The results of the correlation analysis between the original characteristic variables and the six principal components (Table 2) showed that PC1 mainly represented the leaf size, leaf length, leaf width, and leaf area. This principal component can be defined as the size factor of the leaf. PC2 mainly represented the leaf shape and leaf shape index, while PC4 mainly represented the leaf apex and leaf vein logarithm, both of which were related to the leaf shape. These two principal components were defined as the leaf shape factor. PC3 mainly represented the depth and logarithm of the leaf teeth, which are defined as the leaf teeth factors. PC5 mainly represented the leaf texture and was defined as a leaf texture factor. Similarly, PC6 was defined as the vein factor for the leaf vein logarithm.

**Table 2.** Principal component scores of each phenotypic characters measured of 145 samples of ancient tea plant germplasm.

| Variable | Factor Loading | | | | | |
|---|---|---|---|---|---|---|
| | PC1 | PC2 | PC3 | PC4 | PC5 | PC6 |
| Height of tree (m) | 0.555 | 0.208 | −0.495 | −0.118 | 0.222 | 0.096 |
| Tree shape | 0.457 | 0.126 | −0.686 | 0.357 | 0.057 | 0.186 |
| Leaf texture | 0.172 | −0.042 | 0.200 | 0.174 | 0.890 | −0.270 |
| Leaf apex | −0.123 | −0.356 | 0.037 | 0.591 | −0.307 | −0.450 |
| Leaf shape | 0.339 | 0.799 | 0.343 | 0.125 | −0.105 | −0.135 |
| Leaf size | 0.871 | −0.209 | 0.109 | −0.027 | −0.093 | −0.190 |
| Leaf tooth depth | −0.143 | −0.301 | 0.568 | 0.355 | 0.091 | 0.058 |
| Length of leaf (cm) | 0.960 | 0.055 | 0.179 | −0.065 | −0.096 | −0.067 |
| Width of leaf (cm) | 0.859 | −0.467 | −0.056 | −0.078 | −0.040 | −0.024 |
| Leaf shape index | 0.291 | 0.841 | 0.371 | −0.006 | −0.064 | −0.057 |
| Leaf area | 0.946 | −0.231 | 0.081 | −0.086 | −0.076 | −0.064 |
| Leaf veins logarithmic | 0.351 | 0.031 | 0.197 | 0.632 | 0.011 | 0.580 |
| Leaf tooth logarithmic | 0.087 | −0.351 | 0.584 | −0.376 | 0.032 | 0.345 |

The biplot constructed by the first two principal components is presented in Figure 2. While the two axes of the PCA accounted for 42.5% of all the variation, most of the ancient tea plant germplasm grouped was related to geographic origin. The grouping of samples from different districts or regions in the PCA plot points to genetic admixture among the sampled trees.

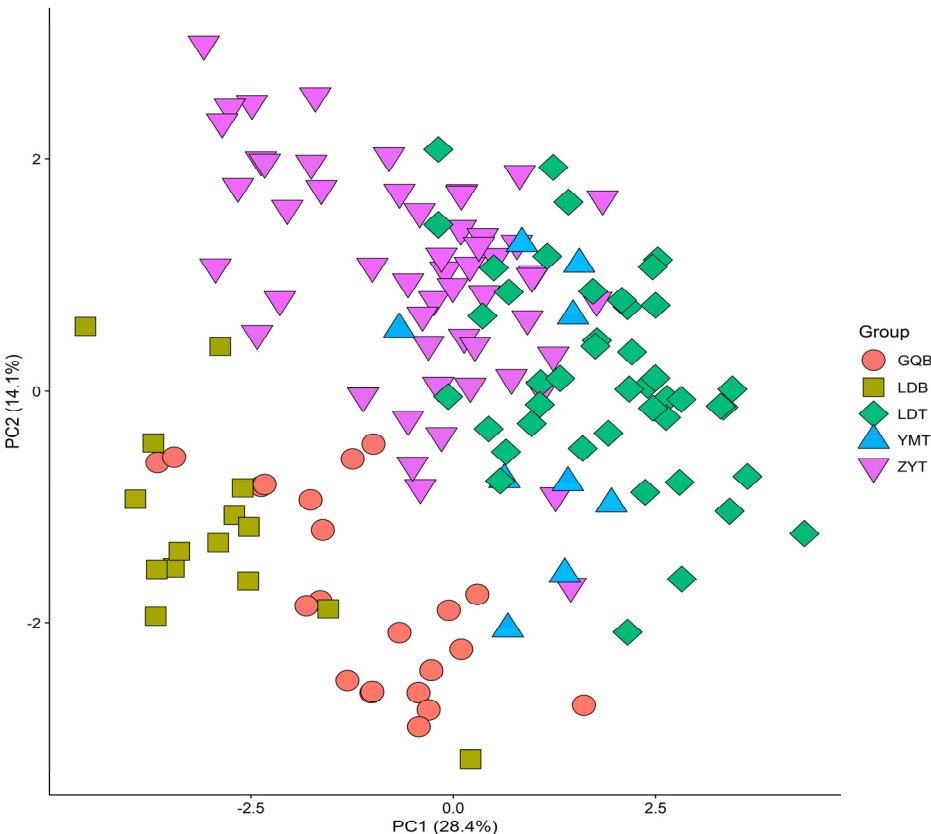

**Figure 2.** Principal component analysis of phenotypic characters measured of 145 samples of ancient tea plant germplasm.

### 3.3. Cluster Analysis of Phenotypic Characters

To conduct the cluster analysis of phenotypic characters, the 145 experimental samples were divided into four groups (Figure S1). The first group comprised 31 tea plant germplasm, which included part of the arboreal ancient tea plant germplasm and all the shrubby ancient tea plant germplasm in Landong village. The second group consisted of 38 tea plant germplasm, composed of part of the arboreal ancient tea plant germplasm from Landong village and all of the ancient tea plant germplasm from Yangmeng village. Ancient tea plant germplasm of a total of 22 shrubs from the Guqi village formed the third group, whereas 54 arboreal ancient tea plant germplasm from Zenya village comprised the fourth group.

### 3.4. SSR Markers Development

In this study, fifteen pairs of primers were successfully identified, indicating highly polymorphic, stable amplification, and interspecific differences among the 145 tea plant germplasm resources (Table 3). The PIC values of the SSR markers ranged from 0.31 to 0.87, with an average of 0.66. All of the PIC values of 15 SSR loci were higher than 0.30, indicating that their effectiveness was in genotype identification.

**Table 3.** Names and sequence information of the 15 SSR markers used for genetic analysis.

| Number | Primer ID | Forward Primer Sequence (5′-3′) | Reverse Primer Sequence (5′-3′) | Repeat Moti | Expected Product Size (bp) | PIC Value |
|---|---|---|---|---|---|---|
| 1 | Cs1 | ATGCCCTCTACATGCCTTTG | ATGACGTAGGCGGAAACAAC | (TC)6 | 179 | 0.48 |
| 2 | Cs2 | GAGCTGAGGCAGTCCATAGG | AAAAGGGAGAAAGACGTGGG | (GT)7 | 217 | 0.74 |
| 3 | Cs3 | GCCATCATAGACTGCTCGGT | GGTTGGCTTGACAAAAAGGA | (TA)10 | 278 | 0.73 |
| 4 | Cs4 | CTCTTCCTCAGCCACCAAAG | TGAGGAGTTGTGGCAGAATG | (AAG)6 | 199 | 0.81 |
| 5 | Cs5 | GGGATCAGATATGGAGCCAA | CAGCAAATTCTTGAGGAAAC | (TTC)5 | 240 | 0.82 |
| 6 | Cs6 | GCTCATTGGCTTTGGCTTT | AATGCATTCCGTAAGCTTGG | (CTA)5 | 180 | 0.31 |
| 7 | Cs7 | GAACAGGCGAACAAGTAGGG | CGACCTCTGAGGCAATCTTC | (GAA)6 | 253 | 0.77 |
| 8 | Cs8 | GGCTTGCATGCCAGTTTATT | ACGTGGGGTTGGAAGACATA | (GT)7 | 195 | 0.87 |
| 9 | Cs9 | CCACGCTTTCTAACACCCAT | AAGGCTCCAAATGCTGAAGA | (TC)6 | 183 | 0.62 |
| 10 | Cs10 | CTGAGTCGGGACAGTTTGGT | CCCAAGAGGTGGAAATAGCA | (CT)6 | 253 | 0.63 |
| 11 | Cs11 | CGTGCAATTGAGAATGCTGT | GGGTCGCTGTCTCTACTTGG | (TC)7 | 222 | 0.62 |
| 12 | Cs12 | AACCATGCAGCAAGACACTG | CCCGTAGGAGGTGCATAAGA | (CAA)5 | 129 | 0.36 |
| 13 | Cs13 | AAAGTGGTCGGTGTCCAAAG | TAACAGGTTTCATCCCTGGC | (GAT)5 | 235 | 0.78 |
| 14 | Cs14 | CTTTTGGCCATTGTCAAGGT | CAGACCTATCGAAAACCCGA | (CAC)6 | 242 | 0.68 |
| 15 | Cs15 | CCCACTCCTAAACTCACCCA | AGCCATCACATTGTCCAACA | (CAC)5 | 254 | 0.65 |

*3.5. Genetic Diversity Analysis of Ancient Tea Plant Germplasm Using SSR Markers*

Genetic concordance and genetic distance of the ancient tea plant germplasm were analyzed in PopGene 32 (Tables S7 and S8) [40]. The ancient tea plant germplasm with a high genetic consistency had a low genetic distance and vice versa. The results showed that the genetic consistency among the 145 samples of the ancient tea plant germplasm was 0.5765–0.9529. The genetic consistency of the arbors was between 0.5882 and 0.9529, and the largest one was for the ancient tea plant germplasm numbered 25 and 37. This indicated that their genetic background was similar. The lowest genetic consistency for the germplasm was numbered 48 and 102, indicating their genetic distance was relatively far. The genetic consistency of the ancient tea plant germplasm of shrubs ranged from 0.6118 to 0.9294. The trees numbered 124 and 126 had the highest genetic consistency, indicating that their genetic backgrounds were very similar. The two shrubs numbered 113 and 132 had the lowest genetic consistency, indicating that their genetic distance was relatively far. The genetic consistency between arbors and shrubs is 0.5765–0.9176. The maximum genetic consistency was observed between the arbor numbered 105 and the shrub numbered 108. The minimum genetic consistency was noted between the arbor numbered 48 and the shrub numbered 127, indicating that the genetic distance between these two tea plant germplasms was relatively farther. The results suggested that the highest value of genetic consistency within the arbors and within the shrubs came from one sampling site, and the ancient tea plant germplasm resources with the lowest genetic consistency were from various sampling sites. The PopGene 32 software was applied to analyze the genetic diversity of the 145 tea plant germplasm resources. The results showed that the average observed allele number (Na) was 2.0000, the average effective allele number (Ne) was 1.3984, the average Nei's genetic diversity index (H) was 0.2584, and the average Shannon information diversity index (I) was 0.4119. The results indicated that the genetic diversity among the 145 tea plant germplasm resources was relatively high.

Fifteen SSR markers were used to study the genetic diversity of five different populations of ancient tea plant germplasm (Table 4), and the results showed that the studied populations have a relatively high diversity. Additionally, the same genetic parameters between the different populations were compared to evaluate the differences between them. The results showed that the allele number (Na), the Nei's genetic diversity index (H), the Shannon information index (I), and the percentage of polymorphism sites (PPB) of the five tea plant populations were the highest for the samples collected from Zenya Village, and the lowest for those from Yangmeng village. Therefore, among the five tea plant germplasm populations, the genetic diversity of the ancient tea plant germplasm population in Zenya Village was the most abundant. The genetic diversity of the ancient tea plant germplasm in the Yangmeng Village population was lower than that observed for other ancient tea plant populations. These results were, to some extent, consistent with

the results of the phenotypic analysis. In addition, the number of effective alleles (Ne) and expected heterozygosity (He) were the highest in Guqi Village, indicating that the genetic diversity in the Guqi Village population was relatively high.

**Table 4.** Genetic diversity parameter analysis for five populations of ancient tea plant germplasm using 15 SSR markers.

| Site ∖ Site | Na | Ne | H | I | PPB (%) | Ho | He |
|---|---|---|---|---|---|---|---|
| Landong (arbor) | 1.9188 | 1.2264 | 0.1595 | 0.2730 | 91.88 | 0.899 | 0.710 |
| Zenya (arbor) | 1.9612 | 1.2238 | 0.1637 | 0.2840 | 96.12 | 0.899 | 0.733 |
| Yangmeng (arbor) | 1.6659 | 1.2283 | 0.1524 | 0.2499 | 66.50 | 0.899 | 0.665 |
| Guqi (bush) | 1.8680 | 1.2329 | 0.1603 | 0.2719 | 86.80 | 1.000 | 0.795 |
| Landong (bush) | 1.7745 | 1.2325 | 0.1591 | 0.2652 | 77.45 | 1.000 | 0.793 |

Na—number of different alleles; Ne—number of effective alleles; H—Nei's genetic diversity index; I—Shannon's information index; PPB—the percentage of polymorphic loci; Ho—observed heterozygosity; He—expected heterozygosity.

### 3.6. Analysis of Molecular Variance (AMOVA)

The results of the analysis of molecular variance are presented in Table 5. The results of AMOVA showed that, of the total genetic variability, 95% variability was due to within-population diversity, and 5% was due to among-population genetic differentiation. In other words, the genetic variation within populations was greater than that among populations.

**Table 5.** Analysis of molecular variance (AMOVA) of the studied populations.

| Sources | Df | SS | MS | Est.Var | % |
|---|---|---|---|---|---|
| Among Populations | 4 | 56.46 | 14.12 | 0.22 | 5.00 |
| Within Populations | 144 | 643.50 | 4.47 | 4.47 | 95.00 |
| Total | 148 | 699.96 | | 4.68 | 100.0 |

Df, degree of freedom; SS, sum of squared observations; MS, mean of squared observations; EV, estimated variance.

### 3.7. Population Structure Analysis

The SSR allelic data were analyzed using the model-based STRUCTURE software. The burn-in period was 10,000, the number of repeats was 1000, the number of iterations was 10, and the *K* was tested for 1–7 [41]. The structure obtained by the Evanno method resulted in the separation of the 145 tea plant germplasm resources into three sub-populations (Figure 3a). Among them, the sub-population-I (blue) includes the arboreal materials of Landong Village. The sub-population-II (red) includes the arbor materials of Zenya Village and Yangmeng Village. The sub-population-III (green) includes the shrubby materials of Guqi Village and Landong Village. Further, summary statistics obtained from the STRUCTURE software, in the form of a color chart, indicated a varying degree of genetic intermixing between the accessions from different regions, as presented in Figure 3b.

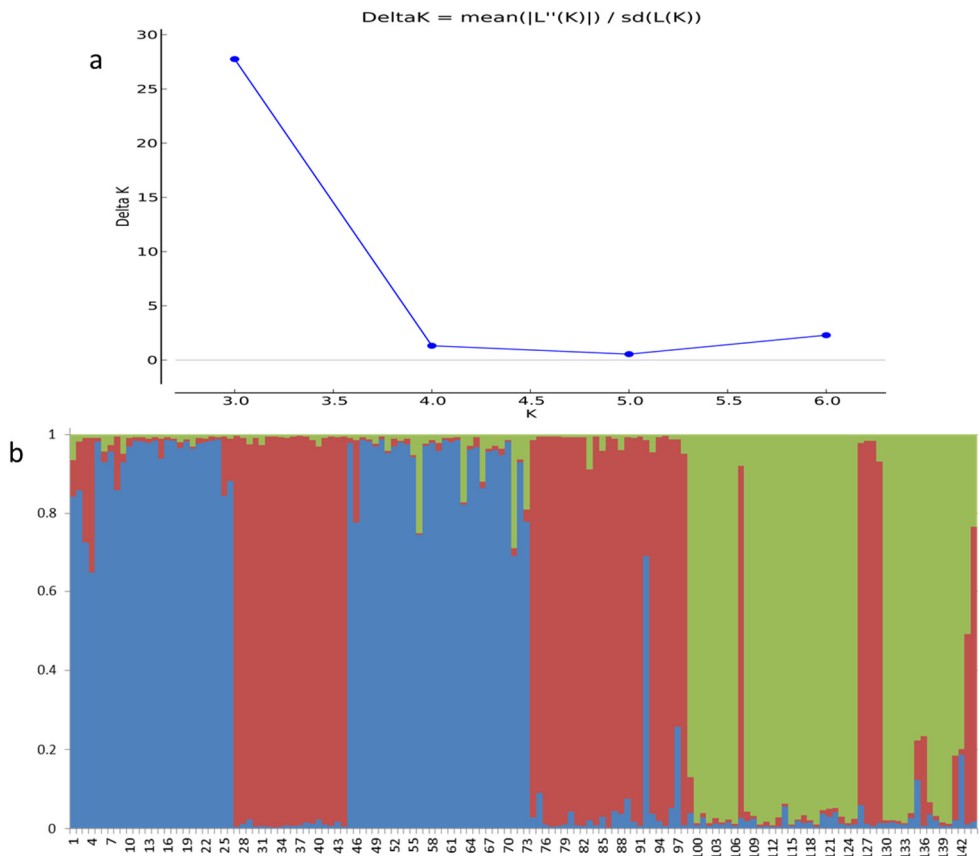

**Figure 3.** Population structure analysis of ancient tea plant germplasm using SSR markers. (**a**). Estimation of the number of populations using LnP (D)-derived Delta k from 2 to 7 using SSR data. (**b**). Model-based population structure analysis of 145 ancient tea plant germplasm.

### 3.8. UPGMA Cluster Analysis Using SSR

Cluster analysis of these 145 ancient tea plant germplasm was carried out with NTSYS-PC 2.1, and the UPGMA tree diagram (Figure S2) was obtained. It was found that there was a genetic similarity coefficient of 0.58, which divided the 145 tea plant germplasm into four groups. The first group included 44 tea plant germplasm of Landong Village (arbor), the second group included 54 tea plant germplasm in Zenya Village, and the third group included nine ancient tea plant germplasm in Yangmeng Village. The fourth group had one individual of the shrub ancient tea plant germplasm in Yangmeng Village, 22 individuals of the shrub ancient tea plant germplasm of Guqi Village, and 15 individuals of the shrub ancient tea plant germplasm of Landong Village. The results from the cluster analysis suggested that the shrub ancient tea plant germplasm and the arboreal ancient tea plant germplasm had a greater genetic distance and far relationship, which were to some extent consistent with the results of the phenotypic clustering results.

## 4. Discussion

The erosion of plant genetic diversity is a serious problem, resulting from both man-made and natural factors [42,43]. The collection and preservation of plant genetic resources are of immense importance for crop breeding, to support the demands of the growing human population. Effective management and utilization of plant genetic resources require information such as phenotypic traits and genetic diversity. Genetic diversity guarantees future genetic progress and protection against climate change, pests, and diseases [44]. It is important to note that ancient and traditional varieties are usually better adapted to the local environmental conditions and diseases [45]. Those varieties include the highly

valuable, yet underutilized, germplasm resources for future breeding. Furthermore, the existence of ancient trees that are presumed to be very old may present a unique genetic window into tea domestication in the world, and therefore, the conservation of the ancient tea plant germplasm resources should be prioritized [46].

The leaf traits of the individuals of the ancient tea plant germplasm in this study were highly diverse. For example, some leaves were small, some were of medium size, some were large or even extra-large, and the leaf shapes were nearly round, oval, long ellipse, or lanceolate. Additionally, there were also differences in the leaf color and leaf apex. In some cases, the differences in the phenotypic traits were related to genetic variation [47]. A greater genetic variation can lead to the abundant phenotypic characters in the ancient tea plant germplasm [48]. Niu [49] analyzed the diversity in ten phenotypic characters of 144 ancient tea plant germplasm resources in Guizhou; it reported that in addition to tree appearance, the variation coefficients of the other nine studied phenotypic characters were higher than 35%. Furthermore, the study reported that the diversity indices of all the ten phenotypic characters were above 0.85. Overall, the values of the variation coefficients and diversity indices in this study were found to be lower than those reported by Niu [49]. This could be because the samples in the previous study were collected from 32 ancient tea plant germplasm resources distributed in Guizhou, whereas this study focused on the phenotypic diversity of the ancient tea plant germplasm in Sandu of Guizhou. The genetic exchange of the ancient tea plant germplasm in the whole province was higher than that at a single region, and the accumulated variation was greater for the phenotypic traits. Our study, which focused on the phenotypic diversity of ancient tea plant germplasm, reported higher phenotypic diversity compared with the results reported by Huang [50] and Xie [51]. They analyzed 30 phenotypic characters; however, Huang sampled 100 Longjing tea plants from ten villages in the Xihu District, Hangzhou, while Xie sampled 109 middle- and small-leaf tea plants in Sichuan. The higher phenotypic diversity in this study could be caused by (1) a longer duration of natural selection, which leads to the accumulation of genetic variation and higher phenotypic diversity [52], and (2) the specific environmental factors, which impact the phenotypic characteristics of the tea plant germplasm [53].

Because of the characters of co-dominance, simple operation, good generality, and high reproducibility [54], SSRs are one of the most prevalent genetic markers, and they play a significant role in plant genetics and breeding. However, the effectiveness and success of SSR markers rely on the quality of the markers, accuracy of the genotyping data, and the plant materials [55]. In this study, we paid more attention to the process of marker selection and selected 15 markers from 90 SSR loci to analyze the population genetic diversity of the ancient tea plant germplasm. Our results showed that these 15 SSR markers were highly polymorphic in 145 samples, with an average effective allele number (Ne) of 1.3984 and an average PIC value of 0.66. The PIC value is an important feature of SSR markers and it is often used to measure the informativeness of a genetic marker for genetic studies. [56]. Remarkably, 13 SSR loci were highly polymorphic (PIC > 0.5), whereas 2 SSR loci showed low PIC values (PIC < 0.5) [57]. The results suggested that the Ne and PIC values were greater than those reported in several previous studies for EST-SSR markers [58].

The population structure analysis and UPGMA clustering tree can reflect the genetic diversity and genetic distances among plant species and/or varieties [59]. The tea shrubs were clustered together, possibly because shrubs have a shorter evolutionary time and lesser genetic variation than arbors; hence, shrubs are closely related to each other [60]. Additionally, the results of the population structure analysis showed that the arbors from Zenya and Landong were clustered together. In comparison, among all the arboreal ancient tea plant germplasm populations, these two arboreal ancient tea plant germplasm populations were closely related. A possible reason could be that human activity causes frequent gene exchange between the two populations. Since the studied populations were surrounded by mountains, and the traffic inconvenience of Sandu of Guizhou could block human-mediated gene exchange between the ancient tea plant germplasm resources located in different regions, we had expected that geographical isolation would promote

interpopulation variability. However, only 5% of the total variability was attributed to variability between the studied populations. This could ultimately lead to genetic stability and aggregation in a particular region. Additionally, the results of the phenotypic trait analysis in this study were slightly different from that of the SSR genetic diversity analysis. A possible reason could be that phenotypic traits are easily affected by environmental factors, and some traits can only be expressed at a specific time.

The average gene diversity (He), which is a measure of genetic diversity observed, was higher compared with the previously genetic diversity reported in the Yunnan province in China [61]. Moreover, the observed heterozygosity of the genotypes was high, which indicated that the ancient tea plant germplasm populations in the Guizhou Province had high polymorphism. Niu et al. [62] found that the Guizhou Plateau has many ancient landraces and pure wild-type accessions, both of which can be used for tea breeding. Therefore, future studies should focus more on the tea germplasm in the Guizhou Plateau.

We analyzed the genetic diversity of the ancient tea plant germplasm through morphological and molecular characterization. It provides a preliminary assessment of the genetic diversity, as well as accurate and critical information for the identification of genetic resources. However, the limitations of the methods used for the phenotypic trait assessment are that (1) this method is easily influenced by environmental factors; and (2) only one molecular marker was used [63]. Traditional experimental screening of the SSR marker polymorphism is laborious and time consuming. Therefore, to better facilitate the conservation and utilization of global tea resources, more robust markers (e.g., SNPs) need to be developed [64].

## 5. Conclusions

This study used phenotypes and SSR markers to analyze the genetic diversity of the ancient tea plant germplasm in the Sandu Aquatic Autonomous County. It can be concluded that there is a considerable degree of phenotypic and genetic diversity in the ancient tea plant germplasm in Sandu County of Guizhou Province in China. This will provide valuable information related to the breeding strategies. Through selecting and breeding, new and more productive varieties can be developed to adapt to the changing environment and climate. In addition, the study found that genetic variation within populations is greater than among populations. Therefore, collecting enough representative individuals within the same population is required to protect the genetic diversity of germplasm resources. At the same time, the germplasm resources of different populations should be collected and preserved to maximize their genetic diversity. Thus, this study provides a theoretical basis for the protection and utilization, and promotes further research of the ancient tea plant germplasm. In addition, different DNA molecular markers can reveal different and complementary information about the same gene. Therefore, screening other types of molecular markers for the genetic diversity analysis of the ancient tea plant germplasm may potentially improve our results.

**Supplementary Materials:** The following are available online at https://www.mdpi.com/article/10.3390/d13060276/s1, Table S1: standardized treatment and assignment of phenotypic characters of the tea plant leaves used for diversity analysis. Table S2: statistical analysis of the seven qualitative characters of 145 ancient tea plant germplasm used for diversity analysis of phenotypic characters. Table S3: statistical analysis of the six quantitative characters of 145 ancient tea plant germplasm used for diversity analysis of phenotypic characters. Table S4: variation coefficients of the phenotypic characters of five populations of ancient tea plant germplasm used for diversity analysis. Table S5: diversity index (H) of phenotypic characters of five populations of ancient tea plant germplasm used for diversity analysis. Table S6: eigenvectors and contribution rates of the principal components of 13 phenotypic characters. Table S7: genetic concordances estimates in the 145 ancient tea plant germplasm by using 15 SSR markers. Table S8: genetic distance estimates in the 145 ancient tea plant germplasm by using 15 SSR markers. Figure S1: cluster diagram of phenotypic traits showing clustering relationship of 145 samples of ancient tea plant germplasm. Figure S2: classification and

identification of 145 samples of ancient tea plant germplasm showing genetic relationship based on 15 SSR markers.

**Author Contributions:** Y.Z., D.-G.Z. and X.D. conceived and planned the experiments. R.W. and Q.L. carried out the experiments. R.W. wrote the manuscript. All authors have read and agreed to the published version of the manuscript.

**Funding:** This study was funded by the Key Laboratory of Mountain Plant Resources Protection and Planting Innovation, Ministry of Education (Guizhou University), Open Project funded by independent Project "Survey, Identification and Utilization of Ancient tea plant germplasm Resources in Sandu"; Science and Technology Support Program (Agriculture) of Guizhou, China. Qian Ke He Zhi Cheng No. [2020]1Y001; Cultivation of high-level innovative talents in Guizhou Province.Qian Ke He Ren Cai No. (2016)4003.

**Institutional Review Board Statement:** Not applicable.

**Informed Consent Statement:** Not applicable.

**Data Availability Statement:** The data that support the findings of this study are available from the corresponding author on reasonable request.

**Acknowledgments:** The authors would like to acknowledge Chen Xianxiang and Pan Xuanhua, teachers from Sandu County Tea Office, for leading us to collect samples of ancient tea germplasm.

**Conflicts of Interest:** No competing or conflict of interests declared by all the authors.

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
