# Peer review of "Genetic Diversity of Ancient Camellia sinensis (L.) O.Kuntze in Sandu County of Guizhou Province in China"

_diversity, doi:10.3390/d13060276_

Round 1

Reviewer 1 Report

Dear authors,

This paper could be of interest to readers of the journal Diversity. However, in my opinion the paper could be significantly improved, and the manuscript requires major revision. List of references should be updated. Please include more papers for genetic diversity of tea germplasma, isolation and characterization of highly polymorphic microsatellites in tea, and morphometric analysis in general. Additional analysis should be performed - details are given in the attached document. In general, the discussion should be improved because a large part of the discussion boils down to repeating the results and comparing data with the research of other authors, especially those relating to genetic analysis. Additional corrections to minor errors and text editing, and comments and suggestions are marked in the attached document.

Kind regards

Reviewer 2 Report

General evaluation

In general, genetic population study in plants provides very valuable data concerning the level of gene diversity and genetic differentiation among populations of a given species. In the present paper, the Authors assessed the neutral 15 microsatellite loci variation in an ancient tea 145 germplasms (Camellia sinensis) in China, which can be useful in future protection measures and utilization of these plants for commercial purposes.

Experimental design

Molecular markers analysis and the parameters chosen to depict the diversity and the differentiation levels need some improvements. I guess the PCR reactions have not been performed in multiplexes, and you did not give details about fluorescence labels of primers F and R of 15 loci (Table 3) which would be useful if someone repeats the study.

Capillary electrophoresis was performed in which apparatus (Producer, country)?

Apart from molecular studies, the morphological assessment of variable traits of leaves and trees has been correctly described.

Validity of the findings

The general approach and obtained findings are well described. Although the SSR-based data, could be more discussed (i.e. AMOVA results, heterozygosity level). Experiments are new and carried out on broadly accepted scientific methodology, being of interest to a different audience.

Comments for the Author

L17 – Please add “polymorphic” as an important feature of the markers identified among 145 samples”

L81 – replace “combined 80 with capillary electrophoresis” by “assessment” because this methodology detail is explained in methods

L83 – replace “be screened and developed” by “have been screened and developed”

L88 – add “characteristics” of the materials….

More comments, suggestions, and advice are enclosed in the pdf file here attached.

I would also recommend an English language revision because many typing errors are still present.

Round 2

Reviewer 1 Report

Dear authors,
I have read revised version of your manuscript several times.

The results of the genetic diversity of studied tea plant germplasm should be better discussed.

Table 4 should be checked. Especially results for the average number of alleles of per locus.

Additional corrections to errors and text editing, and comments and suggestions are marked in the attached document.

Please include more papers for genetic and morphometric diversity of tea plant germplasm.

Forests 2019,10, 780; doi:10.3390/f10090780

Front. Plant Sci., 24 November 2020 | https://doi.org/10.3389/fpls.2020.603819

Niuet al. BMC Plant Biology (2019) 19:328 https://doi.org/10.1186/s12870-019-1917-5

Liuet al. BMC Genomics (2019) 20:935 https://doi.org/10.1186/s12864-019-6347-0

Xia et al. Horticulture Research (2020) 7:7 https://doi.org/10.1038/s41438-019-0225-4

Genet Resour Crop Evol 52, 41–52 (2005). https://doi.org/10.1007/s10722-005-0285-1

J Cell & Plant Sci 2010 1 (1): 13-22

The Journal of Horticultural Science and Biotechnology, 77:6, 729-732, DOI: 10.1080/14620316.2002.11511564

Genetika 48(1):87-96 doi 10.2298/GENSR1601087B

and so on…

Best of luck with the future work!

Round 3

Reviewer 1 Report

Dear authors, the manuscript has been improved.

Additional corrections to minor errors and text editing, and few suggestions are marked in the attached document.

Best of luck with the future work!

Author Response

This manuscript is a resubmission of an earlier submission. The following is a list of the peer review reports and author responses from that submission.

Round 1

Reviewer 1 Report

Dear authors,

I have read your manuscript entitled „Genetic diversity and SSR markers development of ancient Camellia sinensis in Sandu County of Guizhou Province“. Unfortunately, the paper has weaknesses and shortcomings that strongly limit its suitability for publishing in the journal Diversity. In the title, you pointed out the development of new microsatellite markers, while there is not a single sentence about it in the article. Why didn’t you describe the isolation of a set of new genomic microsatellite markers? It is not clear whether these are new markers or some that have been previously developed. In general, the statistical methods used in the analysis of genetic and morphological diversity are outdated and the results are not clearly presented. Furthermore, the Discussion section lists the results that were not analysed in the article. The authors state that the genetic distances are positively correlated with the geographical distances, however, Mantel test between genetic and geographical distances was not performed. In addition, in the Conclusion section authors stated that genetic variation within populations is greater than among populations. Population variability has not been analysed - there are no AMOVA results for genetic data, nor ANOVA for morphometric data.